# Systematic Investigation of Phosphate Decomposition and Soil Fertility Modulation by the Filamentous Fungus *Talaromyces nanjingensis*

**DOI:** 10.3390/microorganisms13071574

**Published:** 2025-07-03

**Authors:** Xiao-Rui Sun, Pu-Sheng Li, Huan Qiao, Wei-Liang Kong, Ya-Hui Wang, Xiao-Qin Wu

**Affiliations:** 1Co-Innovation Center for Sustainable Forestry in Southern China, College of Forestry and Grassland, Nanjing Forestry University, Nanjing 210037, China; 2024057@ynau.edu.cn (X.-R.S.); lps1996127@163.com (P.-S.L.); qh_qiaohuan@163.com (H.Q.); k3170100077@njfu.edu.cn (W.-L.K.); wangyahui25@126.com (Y.-H.W.); 2Yunnan Key Laboratory of Coffee (YAU), College of Tropical Crops, Yunnan Agricultural University, Puer 665099, China

**Keywords:** phosphate-solubilizing microbes, ambient temperature, soil type, P homeostasis, P-use efficiency, environmental adaptability

## Abstract

Phosphate-solubilizing microbes (PSMs) in soil play a crucial role in converting insoluble phosphates into plant-available soluble phosphorus. This paper systematically presents a comprehensive array of qualitative and quantitative techniques to assess the phosphate-decomposing capabilities of microbes. Additionally, it introduces two optimized media, namely improved Monkina medium No. 1 and No. 2, which are particularly suitable for detecting the solubilization abilities of microbes toward insoluble organic phosphates. *Talaromyces nanjingensis*, a novel fungal species recently isolated from the rhizosphere soil of *Pinus massoniana*, demonstrates remarkable phosphate-solubilizing abilities. Across multiple temperature gradients (15 °C, 20 °C, 25 °C, 30 °C, and 37 °C), it effectively decomposes both insoluble inorganic and organic phosphates. This is achieved through the secretion of organic acids, including gluconic acid (6.10 g L^−1^), oxalic acid (0.93 g L^−1^), and malonic acid (0.17 g L^−1^), as well as phosphate-solubilizing enzymes. Moreover, under low-, medium-, and high-temperature conditions, *T. nanjingensis* can decompose insoluble phosphates in three types of soil with varying pH levels, thereby enhancing the overall soil fertility. Genomic analysis of *T. nanjingensis* has identified approximately 308 genes associated with phosphate decomposition and environmental adaptability, validating its superior capabilities and multi-faceted strategies for phosphate mobilization. These findings underscore the wide applicability of *T. nanjingensis* in maintaining soil phosphorus homeostasis and optimizing the phosphorus use efficiency, highlighting its promising potential for agricultural and environmental applications.

## 1. Introduction

Phosphorus is the second most essential plant nutrient after nitrogen. Soluble phosphorus in soil is easily fixed and cannot be absorbed by plants, resulting in nutrient deficiencies. Insoluble P in soil includes inorganic solid P and organic solid P [1]. Inorganic solid P exists mainly in the form of variscite (AlPO_4_·2H_2_O) and strengite (FePO_4_·2H_2_O) in acidic soils, while tricalcium phosphate (Ca_3_(PO4)_2_) exists mainly in neutral and calcareous soils (alkaline) [2]. The most common forms of organic solid P in soil include phytate, polyphosphates, and phosphonates [3]. Inositol hexaphosphate, or phytate, is the most common substance produced when microbes degrade plant residues, reacting with clays and sesquioxide to form insoluble compounds [4]. It forms complexes with iron and aluminum proteins and insoluble salts in acidic soils and with calcium in calcareous and alkaline soils. The organic solid P in soil is mainly composed of high-molecular-weight phosphorus compounds, which account for 61–73% of the soil organic P. Phosphorus in high-molecular-weight phosphorus-containing compounds is linked to the polymer structure through phosphomonoester bonds [5].

Soil microbes are the resident inhabitants of the soil, and the soil microbial community plays an important role in stabilizing and improving the physical and chemical properties of the soil. The decomposition of insoluble phosphates in soil is influenced by phosphate-solubilizing enzymes and acids [6,7]. Most microbes can only secrete organic acids and rarely produce inorganic acids. Organic acids can also promote the decomposition of insoluble potassium-bearing minerals in soil [8]. In addition, acidic (such as acids) and alkaline (such as ammonia) substances secreted by microbes can regulate the pH of the soil. The water-soluble salt in soil is an important index of mineral nutrients that can be quickly used by plants in surface soil, as well as a factor by which to judge whether salt ions in soil limit crop growth. The value of soil EC reflects the conductivity of all water-soluble salts in the soil. Microbes can decompose organic matter and other substances in the soil and conduct ion exchange with the external environment, thus regulating the soil EC. PSMs in soil, including phosphate-solubilizing fungi (PSF) [9,10] and bacteria (PSB) [11,12], play an important role in phosphorus cycling due to their ability to convert insoluble phosphates into soluble phosphorus available to plants. As an important member of rhizosphere growth-promoting fungi, PSF can be developed as biofertilizers to promote plant growth and protect plants from pests and pathogens, thus reducing the use of chemical fertilizers and improving the ecological environment [13].

Organic acids produced by microbes promote the decomposition of inorganic phosphates through reducing the pH value of the rhizosphere and competing with phosphorus for soil adsorption sites; additionally, acidic anions in organic acids can be complexed or chelated with cations such as aluminum, iron, and calcium, which leads to phosphorus precipitation to form soluble complexes [14]. Common organic acids released by PSMs include gluconic acid [15,16], oxalic acid, citric acid [17], lactic acid, tartaric acid, and aspartic acid, etc. PSMs can produce the following enzymes to facilitate phosphorus release from organic compounds: phosphatases (including acid phosphatases, neutral phosphatases, and alkaline phosphatases), phytases, phosphonatases, and C-P lyases. Phosphatase dephosphorylates soil organic phosphate (=organophosphorus) compounds by breaking phosphoester or phoanhydride bonds [18]. Phytases are a class of phosphomonoester hydrolases that can hydrolyze phosphoric acid residues from phytic acid or phytate. Phosphonatases and C-P lyases can promote the decomposition of organic phosphonates by breaking C-P bonds [19].

The decomposition and utilization of insoluble phosphates by microorganisms have been a popular topic in previous studies. In recent years, research on phosphate-solubilizing microorganisms has shown a resurgence, covering aspects such as the isolation and identification of phosphate-solubilizing microorganisms [20,21,22], the relationship between phosphate-solubilizing microorganisms and plant growth [23], and the impact of environmental factors on phosphate-solubilizing microorganisms [24]. This indicates a strong demand for such information in both scientific research and production applications. The fundamental research on phosphate-solubilizing microorganisms (PSMs) decomposing insoluble phosphates is relatively mature, but some complex issues have not been elaborated on in depth. One example is the reason that the ability of microorganisms to decompose on insoluble organic phosphates has always been at a low level (this problem was solved in the improved medium experiment (improved Monkina medium No. 1 and No. 2) in this study and was caused by the low content of lecithin—the organic phosphate raw material used to determine the phosphate-solubilizing ability in the Monkina medium)). In addition, there are also few systematic joint explorations of the biochemical and molecular basis of the microbial phosphate-solubilizing ability or of the application effects in various soils under multi-temperature gradients. This manuscript addresses the following questions: What are the aspects of the decomposition mechanism of microbial phosphate solubilization? Do the phosphate-solubilizing ability and applicability of phosphate-solubilizing microorganisms meet the requirements under different temperature and soil pH conditions? Is the low decomposition ability of microorganisms regarding insoluble organic phosphates due to the insufficient content of organic phosphate substrates (such as lecithin) in traditional media? Can the optimized medium more accurately reflect the true solubilization ability of microorganisms regarding organic phosphates?

Strain JP-NJ4 is a plant-promoting fungus that was selected from the rhizosphere soil of *Pinus massoniana* in our previous study [25,26]. From molecular and phenotypic data, it was identified as a putative novel *Talaromyces* fungal species, designated as *T. nanjingensis* [27]. The preliminary study of our group found that *T. nanjingensis* JP-NJ4 had a strong phosphate-solubilizing ability when cultured at 25 °C [25]. This manuscript explores the above questions and hypotheses by evaluating the phosphate-solubilizing ability of *T. nanjingensis* under different temperature gradients and its applicability in soils with different pH values, as well as revealing the molecular basis of phosphate decomposition in this species. The research results will provide a solid theoretical basis for the practical application and production of phosphate-solubilizing microorganisms (*T. nanjingensis*) from the low-temperature seasons (winter and spring) to the high-temperature seasons (summer and autumn).

## 2. Materials and Methods

### 2.1. Source of Strain

*T. nanjingensis* (strain JP-NJ4) is a phosphate-solubilizing fungus that was isolated from the rhizosphere soil of *P. massoniana* (yellow brown soil) in the Arboretum of Nanjing Forestry University, Nanjing, Jiangsu Province. It was stored in the China Center for Type Culture Collection (CCTCC) (http://cctcc.whu.edu.cn/, accessed on 5 June 2025) with the preservation number M 2012167 (Holotype), in a state of metabolic inactivation under cryopreservation.

### 2.2. Qualitative Determination of Phosphate-Solubilizing Capability in T. nanjingensis

The method of transparent circles is often used to measure the consumption of a target substance. The phosphate-solubilizing capability of *T. nanjingensis* was preliminarily evaluated by using the following insoluble phosphate media. The National Botanical Research Institute’s phosphate medium (NBRIP) is the recommended medium to test the inorganic phosphate-solubilizing capabilities of microbes [28]. This study used 5 formulations of NBRIP medium based on (1) the original NBRIP medium formula—glucose 10 g, (NH_4_)_2_SO_4_ 0.1 g, KCL 0.2 g, MgSO_4_·7H_2_O 0.25 g, MgCl_2_·6H_2_O 5.0 g, Ca_3_(PO_4_)_2_ 5.0 g, agar 20 g, and dd H_2_O 1000 mL. Using the same phosphorus content (1 g L^−1^) (Table 1), 3 types of NBRIP medium were prepared with (2) AlPO_4_, (3) FePO_4_·4H_2_O, and (4) FePO_4_ as the sole phosphorus source (other components were the same as in the original NBRIP medium). (5) Furthermore, an NBRIP medium was prepared with an additional 0.5 g L^−1^ yeast extract powder (YEP) added to the original NBRIP medium (the added amount of YEP (0.5 g L^−1^) referred to that in Pikovskaya medium (PVK)) [28].

Monkina medium (containing an insoluble organic phosphate—lecithin) is the recommended medium to detect the decomposition abilities of microbes regarding insoluble organic phosphates [29]. This study used 3 formulations of Monkina medium: (1) the original Monkina medium—sucrose 10 g, (NH_4_)_2_SO_4_ 0.5 g, MgSO_4_·7H_2_O 0.3 g, NaCl 0.3 g, KCl 0.3 g, FeSO_4_·7H_2_O 0.03 g, MnSO_4_·7H_2_O 0.03 g, CaCO_3_ 5 g, lecithin 0.6 g, agar 20 g, and dd H_2_O 1000 mL; (2) improved Monkina medium No. 1—the added amount of lecithin (the sole phosphorus source) was increased to 5 g L^−1^ and other components were consistent with the original Monkina medium (Table 1); (3) improved Monkina medium No. 2—glucose 10 g, (NH_4_)_2_SO_4_ 0.5 g, MgSO_4_·7H_2_O 0.3 g, NaCl 0.3 g, KCl 0.3 g, FeSO_4_·7H_2_O 0.03 g, MnSO_4_·7H_2_O 0.03 g, CaCO_3_ 1 g, lecithin 0.2 g, agar 20 g, YEP 0.5 g L^−1^, and dd H_2_O 1000 mL.

The inoculation method used in this study was as follows: conidia of *T. nanjingensis* were collected on potato dextrose agar medium (PDA: potato 200 g, dextrose 20 g, agar 20 g, dd H_2_O 1000 mL) at 10 °C and inoculated again on PDA for 30 days at 10 °C. Subsequently, the conidia were washed with sterile deionized water (dd H_2_O) and then diluted with a semisolid agar solution containing 0.2% agar and 0.05% Tween 80 to prepare the pretreated conidia suspension (stored at 4 °C for standby use). The volume of sterilized medium added in each Petri dish was 20 mL. The center of each Petri dish was inoculated with 1 μL of conidia suspension (1 × 10^4^ cfu/μL), and they were cultured at 10 °C, 15 °C, 20 °C, 25 °C, 30 °C, and 35 °C, respectively. Every treatment included 3 biological replicates (Here and in subsequent manuscripts, the number of biological replicates for the samples is abbreviated as n. n = 3). The colony growth and the emergence of a phosphate-solubilizing transparent circle in *T. nanjingensis* after 7 days were observed.

### 2.3. Quantitative Determination of Inorganic and Organic Phosphate-Solubilizing Capabilities in T. nanjingensis Under Different Temperatures

(1) NBRIP liquid medium with Ca_3_(PO_4_)_2_ as the sole phosphorus source was used to determine the decomposition ability of microbes regarding insoluble inorganic phosphates; (2) improved Monkina liquid medium No. 1 and (3) improved Monkina liquid medium No. 3 (with 5 g L^−1^ calcium phytate instead of lecithin as the sole phosphorus source) were used to quantitatively determine the decomposition ability of microbes regarding insoluble organic phosphates. The above liquid media did not contain agar, and the remaining components were the same as in the original solid medium (Table 1).

A single colony plug (8 mm, in a 3.5 cm radius within the medium) of *T. nanjingensis* was taken from the PDA (30 days at 10 °C) and inoculated in a shake flask (volume 50 mL) with sterilized liquid medium (25 mL). The inoculated shake flasks were oscillated at 180 rpm for 7 days at 15 °C, 20 °C, 25 °C, 30 °C, and 37 °C, respectively. The soluble phosphorus content (ascorbic acid–molybdenum blue method (molybdenum–antimony spectrophotometry method, Mo-Sb-Vc)) [30] and pH value (pH meter directly) of the fermentation broth were determined every 24 h for 7 days (n = 3).

### 2.4. Determination of Titratable Acid Production and Acid Composition in Fermentation Broth of T. nanjingensis

The production of titratable acids was determined by acid–base neutralization titration (n = 3). The NBRIP liquid medium with Ca_3_(PO_4_)_2_ as the sole phosphorus source was selected and oscillated at 180 rpm under 15 °C, 20 °C, 25 °C, 30 °C, and 37 °C, respectively. After 3 days, the fermentation broth was centrifuged at 10,000 rpm for 10 min, and 10 mL of the supernatant was taken for determination. The inoculation method was the same as that in Section 2.3.

The organic acids were identified and quantified using high-performance liquid chromatography (Agilent 1200 liquid chromatograph, Santa Clara, CA, USA). The procedures referred to Zeng and Qiao [31,32]. A single colony plug from PDA was inoculated in potato dextrose broth (PDB) and oscillated at 25 °C and 180 rpm for 4 days to obtain the seed solution. The preparation and inoculation methods were the same as in Section 2.3. Specifically, 1 mL seed solution was added to a 250 mL shake flask containing 50 mL NBRIP liquid medium and oscillated at 25 °C and 180 rpm for 5 days (n = 3). One separate sample was taken from each independent culture and mixed for the determination of the acid composition.

### 2.5. Detection of Phosphate-Solubilizing Enzyme Activity in Fermentation Broth of T. nanjingensis

The activity of phosphate-solubilizing enzymes was detected in Monkina medium supplemented with 2.5 g each of lecithin and calcium phytate as a mixed insoluble organic phosphate source. The inoculation method and culture conditions were the same as in Section 2.3. Enzyme activity was determined after 5 days of culture (n = 3). The concentration unit (IUmL^−1^ = μmol/min/mL) of enzyme activity in the fermentation broth of *T. nanjingensis* determined by colorimetry was defined as follows. For ACP, the production of 1 μmol 4-nitrophenol from the substrate catalyzed per mL of fermentation broth per minute at 30 °C was defined as one concentration unit of enzyme activity (1 IUmL^−1^) [33]. For NP and AKP, the production of 1 μmol phenol from the substrate catalyzed per mL of fermentation broth per minute at 37 °C was defined as 1 IUmL^−1^ [33]. For phytase, the production of 1 μmol inorganic phosphorus from 5 mmol/mL sodium phytate turbid solution catalyzed per mL of fermentation broth per minute at 37 °C and pH 5.5 was defined as 1 IUmL^−1^ [34].

### 2.6. Determination of Main Chemical and Physical Properties of Different Soils After Addition of T. nanjingensis

Soil sampling was conducted on 15 June 2021 at 3 locations in Jiangsu Province, China: (1) the manor fields in Pizhou, Xuzhou City and (2) the grasslands and (3) arboretum on the campus of Nanjing Forestry University in Nanjing City (Table 2). Each sampling location contained 3 sampling sites, with 3 replicates in each sampling site (three-point sampling method, 3 × 3). Digging was performed at a depth of 20–30 cm to collect soil samples (3 × (3 × 3)), which were placed in sterile polyethylene bags and stored at 4 °C. After removing debris and air drying, they were sieved (2 mm) and then mixed separately with a grinder. The chemical and physical properties of the original soil were determined: available nitrogen (alkali-hydrolyzable nitrogen, AN), total phosphorus (TP), available phosphorus (AP), available potassium = available kalium (AK), humus (total carbon of humus, HM), pondus hydrogenii (pH), and electrical conductivity (EC) [35]. Except for AN and HM, all indices were determined 3 times.

The protocol to investigate the effects of the addition of *T. nanjingensis* on 3 types of soil with different pH values at different temperatures was as follows. All 3 soil samples were treated in 4 ways: (1) non-sterilization soil + sterile deionized water; (2) non-sterilization soil + conidia suspension; (3) sterilization soil + sterile deionized water; (4) sterilization soil + conidia suspension. Soil application tests were performed in Petri dishes with 80 g soil and 10 mL sterile deionized water or conidia suspensions evenly pipetted into each Petri dish. The above treatment groups were then placed at 10 °C, 20 °C, and 30 °C, respectively (n = 3). Refer to Section 2.2 for the preparation process of the conidia suspension (1 × 10^7^ cfu/mL); 10 mL of sterile deionized water contained the same amount of semisolid agar as 10 mL of conidia suspension.

After 4 weeks, the TP, AP, AK, pH, and EC of the soil were detected. In order to ensure the accuracy and correctness of the experiment, the traditional method of TP, AP, and AK index determination after long-term air drying was improved. The soil treated for 4 weeks was dried at 70 °C to quickly stop the effects of *T. nanjingensis* and other soil microbes on the soil chemical and physical properties. The original non-sterilization/sterilization soils were used as the initial control group (CK) for non-sterilization/sterilization treatments, respectively, and the uninoculated and inoculated conidia suspension groups were mutually controlled (n = 3).

### 2.7. Gene Extraction for Phosphate Decomposition and Environmental Adaptability in the Genome of T. nanjingensis

Second- (Illumina NovaSeq 6000 or DNBSEQ-T7 (Illumina, San Diego, CA, USA)) and third-generation genome sequencing techniques (PromethION P48 sequenator (Oxford Nanopore Technologies, ONT, Oxford, UK)) were used for *T. nanjingensis*. The strain samples required for genome sequencing were obtained through a PDB liquid enrichment culture (with a minimum sampling mass of 0.5 g/tube), and 3 copies were taken and stored in an −80 °C ultra-low-temperature refrigerator. High-quality genomic DNA was extracted using the improved cetyltrimethylammonium bromide (CTAB) method. The above technologies were supported by Wuhan Benagen Tech Solutions Company Limited (Wuhan, China) (http://www.benagen.com/, accessed on 5 June 2025).

In this study, further information mining was carried out on the genomic annotations from 10 platforms (KEGG, KEGG Pathway, Nr, Interpro, Refseq, Pfam, Tigerfam, Uniprot, GO, KOG). Based on the classification and collation of the information for all gene annotation entries obtained from genome sequencing, gene annotation entries related to phosphate decomposition and environmental adaptability in the genome of *T. nanjingensis* were searched for and extracted.

### 2.8. Statistical Analyses

The correlation plot (Version 1.31) and APP modules in OriginPro (Version 2023, OriginLab Corporation, Northampton, MA, USA) were used for correlation analysis, as described in Section 3.2. The correlation type was selected as Pearson, and listwise was selected to exclude missing values in the input data. The correlation coefficient ranged from −1 to 1. Significant data are marked with * for *p* ≤ 0.05, ** for *p* ≤ 0.01, and *** for *p* ≤ 0.001. Microsoft Excel 2016 and SPSS (Version 18.0. IBM, New York, NY, USA) were used to collate and analyze the data (mean ± standard deviation (s.d.)). One-way analysis of variance (ANOVA) and Tukey’s multiple comparisons were employed to assess differences. Significant (*p* < 0.05) and highly significant (*p* < 0.01) differences between different treatment groups are identified by lowercase letters (a, b, c, d, e, f, etc.) and uppercase letters (A, B, C, D, E, F, etc.), respectively. In particular, for AP (see Section 3.5), an additional two-samples T test was introduced (* for *p* < 0.05, ** for *p* < 0.01). The indicated letters used for the non-sterilization group data are bolded, and the original font is used for the sterilization group.

## 3. Results

### 3.1. Qualitative Results Regarding Decomposition of Insoluble Phosphates by T. nanjingensis

The qualitative determination results showed that *T. nanjingensis* could decompose both insoluble inorganic and organic phosphates (Figure 1). In the NBRIP medium supplemented with three inorganic phosphates, Ca_3_(PO_4_)_2_, AlPO_4_, and FePO_4_, transparent circles appeared (Figure 1A). In the original NBRIP medium with YEP, the transparent circle was more obvious and the strain grew faster. In the original Monkina medium (lecithin 0.6 g L^−1^), improved Monkina medium No. 1 (lecithin 5 g L^−1^), and improved Monkina medium No. 2 (lecithin 0.2 g, YEP 0.5 g L^−1^) containing organic phosphates, transparent circles also appeared (Figure 1B). Among them, the transparent circle was the most obvious in improved Monkina medium No. 2 with YEP.

### 3.2. Characteristics of Inorganic and Organic Phosphate Solubilization in T. nanjingensis Under Different Temperatures

Under different temperature gradients, the addition of *T. nanjingensis* could decompose insoluble inorganic phosphates (Ca_3_(PO_4_)_2_) and organic phosphates (lecithin and calcium phytate). With an increase in the culture temperature and time, the soluble phosphorus content in the fermentation broth increased and the pH value decreased (Figure 2A–C). The peak values of soluble phosphorus in the NBRIP liquid medium, improved Monkina medium No. 1, and improved Monkina medium No. 3 were 12.889 ± 0.847 mg/L (30 °C, day 7), 7.357 ± 0.111 mg/L (37 °C, day 7), and 13.403 ± 0.292 mg/L (37 °C, day 4), respectively. The lowest pH values of the three media were 3.91 ± 0.065 mg/L (15 °C, day 7), 2.883 ± 0.062 mg/L (15 °C, day 7), and 3.317 ± 0.074 mg/L (37 °C, day 7), respectively. The results regarding the soluble phosphorus content recorded visually (Figure 2D) were consistent with those reflected in the line graphs (Figure 2A–C).

A pH decrease significantly promoted insoluble phosphate decomposition and increased the soluble phosphorus yield (Figure 2E). The soluble phosphorus content in the NBRIP liquid medium (Ca_3_(PO_4_)_2_), improved Monkina medium No. 1 (lecithin), and improved Monkina medium No. 3 (calcium phytate) was negatively correlated with the pH. The Pearson correlation coefficients were −0.65 (*p* ≤ 0.001, ***), −0.24 (*p* ≤ 0.05, *), and −0.82 (*p* ≤ 0.001, ***), respectively.

### 3.3. Titratable Acid, Acid Composition, and Phosphate-Solubilizing Enzyme Activity in Fermentation Broth of T. nanjingensis

*T. nanjingensis* could produce acidic materials at all five temperature gradients (Figure 3A). In the 15 °C (c), 20 °C (b), and 25 °C (a) treatment groups, the titratable acid content of the fermentation broth showed a rapid increase with the increase in the temperature. Starting at 25 °C, it became stable, and it did not change significantly at 25 °C (a), 30 °C (a), or 37 °C (a) (peak value of 33.33 ± 1.247 mmol/L at 37 °C; that of CK was 1.2 ± 0.156 mmol/L (c)). The total amount of organic acids produced by *T. nanjingensis* was 7.20 g L^−1^, including gluconic acid (6.10 g/L), oxalic acid (0.93 g/L), and malonic acid (0.17 g/L). No lactic acid was detected.

With the increase in temperature, the ACP activity of the fermentation broth increased gradually (Figure 3B) (peak value of 0.392 ± 0.009 μmol/min/mL at 37 °C), and the activity of NP and AKP showed a trend of first decreasing and then increasing. At 30 °C, the activity of NP (0.033 ± 0.004 μmol/min/mL) and AKP (1.936 ± 0.284 μmol/min/mL) reached the lowest values among the five temperature gradients. The trend of phytase activity was the opposite to that of NP and AKP (peak value of 0.057 ± 0.057 μmol/min/mL at 30 °C).

### 3.4. Basic Properties of Soils with Three pH Values

According to the classification criterion of the soil pH, the manor fields, grasslands, and arboretum were alkaline, neutral, and acidic, respectively. The appearance, especially the color, of the three soils was different (Figure 4A). There were significant differences in the TP, AK, pH, and EC among the three soils (Table 2). The AP content in the manor fields and grasslands was similar, and it was significantly different from that in the arboretum. Both the acidic (arboretum) and alkaline (manor fields) soils contained less HM than the neutral (grassland) soil, with the most severe HM loss in alkaline soil (which also had the lowest AN and high EC). The soil of the arboretum was acidic, with low values of TP, AP, AK, and EC. The indices of the grassland were superior. Overall, it was rich in nutrients and more suitable for plant growth.

### 3.5. Effects of T. nanjingensis on Three Types of Soil with Different pH Values

The effects of *T. nanjingensis* on the available phosphorus (AP) content of the soils were the focus of this study (Figure 4B). After 4 weeks of culture, the addition of the conidia suspension increased the AP content in all groups except the 10 °C group with neutral soil. The results of the two-sample T test of the AP index between groups supplemented with the conidia suspension and sterile deionized water indicated that, in the three types of soil at 30 °C and in the acidic soil at 20 °C, the addition of the conidia suspension significantly increased the AP content in the non-sterilization and sterilization treatment groups (*p* < 0.01, **). In acidic soils at 10 °C, *p* < 0.05, *. In the non-sterilization treatment groups of alkaline soil at 10 °C and 20 °C, the addition of the conidia suspension significantly increased the AP content (*p* < 0.05, *).

After 4 weeks of culture, there were no significant differences (*p* > 0.05) between most treatment groups and the soil control groups regarding TP (Appendix A). Highly significant differences (*p* < 0.01) were observed between the + sterile deionized water and + conidia suspension treatment groups at 10 °C and 20 °C regarding the sterilization treatment, but the error was suspected to have been caused by soil sample heterogeneity. The difference in TP content between the two control groups (CK) of non-sterilization soil and the sterilization soil cultured for zero days was small. Organic acids can also decompose potassium ore (insoluble potassium-bearing minerals) in the soil into AK. Regarding AK, in most temperature treatments of the three soils, there was no significant difference between the conidia suspension addition group and the non-addition group, regardless of the sterilization treatment (Appendix A).

In terms of the soil acid–base environment, the addition of *T. nanjingensis* conidia had a complicated effect on the soil pH (Appendix A). The pH of the neutral soil used in this experiment was greater than 7; it was close to alkaline after the sterilization treatment. This might have been due to the decomposition of certain compounds in the soil as a result of sterilization. In addition, the pH values of the two treatments in the non- sterilization group at 30 °C were lower than those of the CK group (the original soil), which might imply that there were some microbes with a strong ability to secrete acidic substances in the original soil. After exogenous water supplementation and constant-temperature culture at 30 °C, the activity of these microbes was enhanced, and a large number of acidic substances were secreted. This also explained the high content of AK in the acidic soil at 30 °C, compared with the respective CK (Appendix A). All water-soluble salts in the soil affected the EC. Overall, under the action of soil-resident microbes and conidia from *T. nanjingensis*, the high EC value of the soil was reduced, and the low EC value was increased (Appendix A).

### 3.6. Genes Associated with Phosphate Decomposition and Environmental Adaptability Found in T. nanjingensis

Genes associated with phosphate decomposition (Table 3 and Appendix A) and environmental adaptability (Table 3 and Appendix A) were found in *T. nanjingensis*. These 308 genes (or items) were phosphate-solubilizing (111)—namely toward organic acids (9), acid phosphatases (29), alkaline phosphatases (28), phytase (7), phosphonatases (3), and C-P lyases (35)—and phosphate-related (197)—namely regarding phospholipases (90), dolichyldiphosphatase (1), pyrophosphatase (18), diphosphatases (15), phosphoesterases (36), and phosphodiesterases (37). In addition, there were 26 genes related to calcineurin-like phosphoesterases. Genes associated with environmental adaptability were involved in aspects such as temperature, salt, and stress resistance and interactions with plants and microbes (siderophores, hormones, salicylic acid/salicylate, and polyamines).

## 4. Discussion

### 4.1. Multiple Detection Media Necessary for Rapid, Comprehensive Screening of Phosphate-Solubilizing Microbes

*T. nanjingensis* exhibited varying degrees of phosphate decomposition in the original and improved qualitative media used to test the ability of microbes to decompose insoluble inorganic and organic phosphates. In the use of traditional detection media, if certain microbes are not detected as having the ability to decompose insoluble phosphates, this does not necessarily mean that they do not have this function. Meanwhile, in the qualitative detection and screening of a large number of phosphate-solubilizing microbes in some samples (such as soil), a large number of qualified members may be missed. In this context, multiple types of screening media are urgently needed.

Under high content or concentrations of lecithin, *T. nanjingensis* showed a stronger ability to decompose phosphates. This is consistent with the conclusion that the phosphate decomposition capacity of some microbes is induced by high phosphorus concentrations and inhibited by low phosphorus concentrations [36]. High substrate content might stimulate the opening and incremental expression of phosphate decomposition pathways in microbes. In addition, YEP accelerated the decomposition of lecithin by *T. nanjingensis*. The phosphate decomposition ability testing medium containing YEP has been used by Nautiyal [28].

### 4.2. Insufficient Lecithin Results in Low Detection of Organic Phosphate Decomposition by Microbes

The original Monkina liquid medium, when freshly prepared, was murky. After inoculating *T. nanjingensis* for only 24 h, the liquid medium in the 25 °C, 30 °C, and 37 °C treatment groups became completely clear (Figure 5). The liquid media of the 15 °C and 20 °C treatment groups also became clear rapidly over the next 2–3 days. In general, the insoluble phosphate liquid medium became completely clear, which meant that the insoluble phosphates were completely decomposed by the strain, and soluble phosphorus appeared. However, after testing, it was found that the content of soluble phosphorus in the clarified liquid medium was almost zero. The above phenomena indicate that *T. nanjingensis* has a strong ability to decompose insoluble phosphates (lecithin), and it also has a strong ability to absorb soluble phosphorus. This means that the amount of lecithin in the original Monkina medium was insufficient. The content of lecithin in the original Monkina medium was 0.6 g L^−1^ (Table 1). With reference to the content of Ca_3_(PO_4_)_2_ in NBRIP medium (5 g L^−1^), the content of lecithin in the medium was increased from 0.6 g L^−1^ to 5 g L^−1^, and the improved Monkina medium No. 1 with increased lecithin content was prepared. The experimental results showed that, after inoculating *T. nanjingensis* in improved Monkina medium No. 1, soluble phosphorus was detected successfully from the culture medium. This medium was more suitable for the determination of the insoluble organic phosphate decomposition ability of microbes with strong phosphate decomposition abilities.

During growth and reproduction, *T. nanjingensis* absorbed soluble phosphorus from lecithin. This behavior was expected. The decomposition of phosphates by microbes is accompanied by the absorption and utilization of soluble phosphorus [37,38]. The function of decomposing insoluble phosphates in such microbes also enables the survival of the strain itself in harsh environments. However, this does not mean that the application of *T. nanjingensis* in soil is harmful or not beneficial to plants. In actual applications, the use of *T. nanjingensis* in soil would not deplete the rhizosphere of soluble phosphorus. One reason is that there is a large quantity of insoluble phosphates in the soil, and the other reason is that the mycelium would also die, and the soluble phosphorus in the mycelium would be released and returned to the rhizosphere soil of the plant. PSMs (such as *T. nanjingensis*) acts as a living reservoir of soluble phosphorus [39], maintaining the content of soluble phosphorus in soils. In addition, during the fermentation culture, the rise in the soluble phosphorus content in the fermentation broth slowed down in some stages (Figure 2A–C), and some treatment groups even showed a decreasing trend (Figure 2A,C). This reflects the phosphorus consumption demands of *T. nanjingensis* during growth and reproduction and also demonstrates its strong ability to absorb and reserve soluble phosphorus.

### 4.3. Determination of Amount of Insoluble Organic Phosphate Added to Improved Monkina Medium

The Monkina medium supplemented with lecithin is usually used for the routine screening of insoluble organic phosphate-solubilizing microbes and for phosphate-solubilizing ability tests. Lecithin is one of the representatives of insoluble organic phosphates [40]. Calcium phytate and other phytates are also insoluble organic phosphate sources in soil [41]. In order to screen for microbes with stronger phosphate decomposition abilities and determine their phosphate decomposition abilities, in this study, the original Monkina medium was improved. Improved Monkina medium No. 1 was prepared by increasing the amount of lecithin in the original Monkina medium (Table 1). Meanwhile, improved Monkina medium No. 3 was prepared by replacing lecithin with calcium phytate.

In 1 L of NBRIP liquid medium, the supplemental amount of calcium phosphate was 5.0 g L^−1^. The relative molecular mass of Ca_3_(PO_4_)_2_ is 310.177, and the relative atomic mass of the P element is 30.974. The proportion of the P element in Ca_3_(PO_4_)_2_ = 61.948/310.177 = 1/5.007. Therefore, the mass of phosphorus elements from calcium phosphate in 1 L of NBRIP medium was (61.948/310.177) × 5.0 g = 0.999 g. The situation regarding the amount of lecithin added to improved Monkina medium No. 1 is slightly more complicated. With the addition of calcium phosphate (5 g L^−1^) in the insoluble inorganic phosphate medium, NBRIP, as a reference, we sought to increase the lecithin content in the original Monkina medium from 0.6 g L^−1^ to 5 g L^−1^. This ensured that the amount of insoluble phosphate added per liter of medium was the same, i.e., 5 g. There are many types of lecithin, with common ones being egg yolk lecithin [42] and soybean lecithin [43]. The molecular formulas and molecular weights of soybean lecithin and egg yolk lecithin are approximately the same. The lecithin used in the present study was egg yolk lecithin (biochemical reagent, BR). (Note: in the reagent, the phosphorus content was greater than 3%, i.e., greater than 0.15 g/5 g.) For egg yolk lecithin, the molecular formula is C_42_H_80_NO_8_P, the CAS No. is 97281-47-5, and the relative molecular weight is 758.060. The P element ratio in C_42_H_80_NO_8_P = 30.974/758.060 = 1/24.474. In 1 L of improved Monkina medium No. 1, the phosphorus content from lecithin was (30.974/758.060) * 5.0 g = 0.204 g, which was consistent with the phosphorus content of the reagent being greater than 0.15 g/5 g. The content of lecithin in improved Monkina medium No. 1 was 8.33 (5.0/0.6 = 8.33) times higher than that in the original Monkina medium.

The molecular formula of calcium phytate is C_6_H_6_Ca_6_O_24_P_6_, the CAS No. is 7776-28-5, and the relative molecular weight is 888.408 (Table 1). The P element ratio in C_6_H_6_Ca_6_O_24_P_6_ = 185.844/888.408 = 1/4.780. Here, 5.0 g of lecithin in the Monkina medium was replaced with 5.0 g of calcium phytate, resulting in improved Monkina medium No. 3. In 1 L of improved Monkina medium No. 3, the mass of phosphorus from calcium phytate was (185.844/888.408) × 5.0 g = 1.046 g. Moreover, the quality of the phosphorus in improved Monkina medium No. 3 was almost the same as that in the NBRIP medium. The addition of 5.0 g L^−1^ ensured the quality and consistency of the insoluble phosphate and phosphorus element in the NBRIP medium (insoluble inorganic phosphate medium) and the improved Monkina medium No. 3 (insoluble organic phosphate medium). This is highly beneficial. Therefore, it is reasonable and appropriate to apply these two media in comparative studies. Finally, the added amounts of insoluble phosphate in the two improved Monkina media were set to 5.0 g L^−1^.

### 4.4. Phosphate Decomposition Strategy of T. nanjingensis

Strong organic acids produced by microbes can decompose insoluble inorganic phosphates [44]. Oxalic acid is more acidic than phosphoric acid [45]. *T. nanjingensis* is also capable of secreting oxalic acid. Compared with organic phosphates, the chemical structure of inorganic phosphates is relatively simple. The chemical structures of Ca_3_(PO_4_)_2_, AlPO_4_, and FePO_4_ are generally similar. The negative correlation between soluble phosphorus and pH clearly indicates that acidic substances contribute to the decomposition of insoluble inorganic phosphates [46].

Enzymes act as catalysts, participating in chemical reactions without being consumed or reduced in content. When the total amount of insoluble phosphates was fixed in the closed culture system, the enzyme content that gradually accumulated in the fermentation broth was sufficient for the substrate, even at low temperatures. Therefore, in the later stage of the fermentation culture, the total amount of soluble phosphorus in the low-temperature groups was closer to that in the high-temperature groups. The decomposition of insoluble organic phosphates was mainly affected by phosphatases, and there was little evidence that organic acids were also directly involved. However, under acidic conditions, some phosphatases (such as ACP, phytase) can perform their enzymatic functions better [47]. *T. nanjingensis* could produce considerable amounts of phosphatase under various temperature gradients. The liquid medium was also supplemented with lecithin and calcium phytate as a mixed insoluble phosphate source, which was more accurate in determining the activity of the four phosphatases. Acidic, alkaline, and neutral phosphatases function best under acidic, alkaline, and neutral conditions, respectively, and they could be accurately detected [48,49,50].

### 4.5. PSF Can Regulate Physical and Chemical Properties of Soil

On the whole, *T. nanjingensis* had the function of regulating the physical and chemical properties of the soil (Figure 4 and Appendix A). In a closed system, the TP content of the soil should be fixed [51]. The PSF (*T. nanjingensis*) substantially reduced P fixation in the soil. In all three soils, the addition of *T. nanjingensis* increased the AP content in both non-sterilization and sterilization soils at three temperatures (although this was not obvious in neutral soils treated at 10 °C). The reason that the conidia had no significant effect on the soil AK content might be that the organic acids secreted by the conidia after germination contributed less to the decomposition of potassium ore. In the sterilization treatment groups at 10 °C, the addition of conidia decreased the EC values of the alkaline and acidic soils. This might mean that the conidia of *T. nanjingensis* survived under pure culture conditions compared to natural conditions, occupied an advantageous ecological niche under low-temperature conditions, and utilized water-soluble salts in the soil [52,53]. In some treatment groups, conidia addition had no effect on the EC. This might indicate that the regulation of the soil EC by the microbes had reached a limit [54]. Titratable acids secreted by microbes play an important role in soil acidification [55,56]. However, the effect of the conidia suspension on soil acidification was unstable. The possible reasons are as follows: (1) acids derived from the conidia suspensions reacted with chemicals in the soil microenvironment and were consumed; (2) only conidia suspensions were used in the application protocol, so the acids secreted might not have been sufficient. The choice to add only the conidia suspension was intended to reduce systematic errors by excluding the influence of fermentation products on the soil. Of course, in practical applications, microbes and their fermentation broth could be directly applied for better effects [57]. In addition, the conidia of *T. nanjingensis* remained active after 6 months in PDA, indicating good storage stability and high potential for this species. *T. nanjingensis* isolated from the rhizosphere soil of *P. massoniana* also showed good soil adaptability. This ensures good survival and colonization in the soil environment, as well as full communication with plants [58].

### 4.6. Multifaceted Evidence of Phosphate Decomposition and Environmental Adaptability

In the genome of *T. nanjingensis*, multiple and abundant genes related to phosphate solubilization were found, which confirms that this fungus has strong, multifaceted phosphate decomposition abilities [31,59]. This also means that *T. nanjingensis* can cope with a variety of P deficiency conditions and can deal with soil P fixation in a diversified way. The fungus also has many genes related to environmental adaptability, indicating its good ability to survive and compete in nature.

## 5. Conclusions

This manuscript systematically describes the decomposition ability of a newly identified, potent phosphate-solubilizing fungus (PSF) toward various insoluble phosphates under laboratory conditions. It also highlights the decomposition capacity of this fungus regarding insoluble phosphorus in three types of soils across natural/sterilized soil environments and multiple temperature gradients, while also considering the biochemical and molecular basis of phosphate solubilization in this strain. Limitations during the experiment included the testing of soils with the same pH but of different types, as well as the prolonged effects of the PSF applied to the soils. Challenges in this research direction include how to use microorganisms to homogenize soil indicators across large-area plots and address the issue of soil heterogeneity. Future research can be directed toward targeting different soil types with specialized PSF, as well as exploring the optimal adaptability between specific plants and their endemic PSF.

## Figures and Tables

**Figure 1 microorganisms-13-01574-f001:**
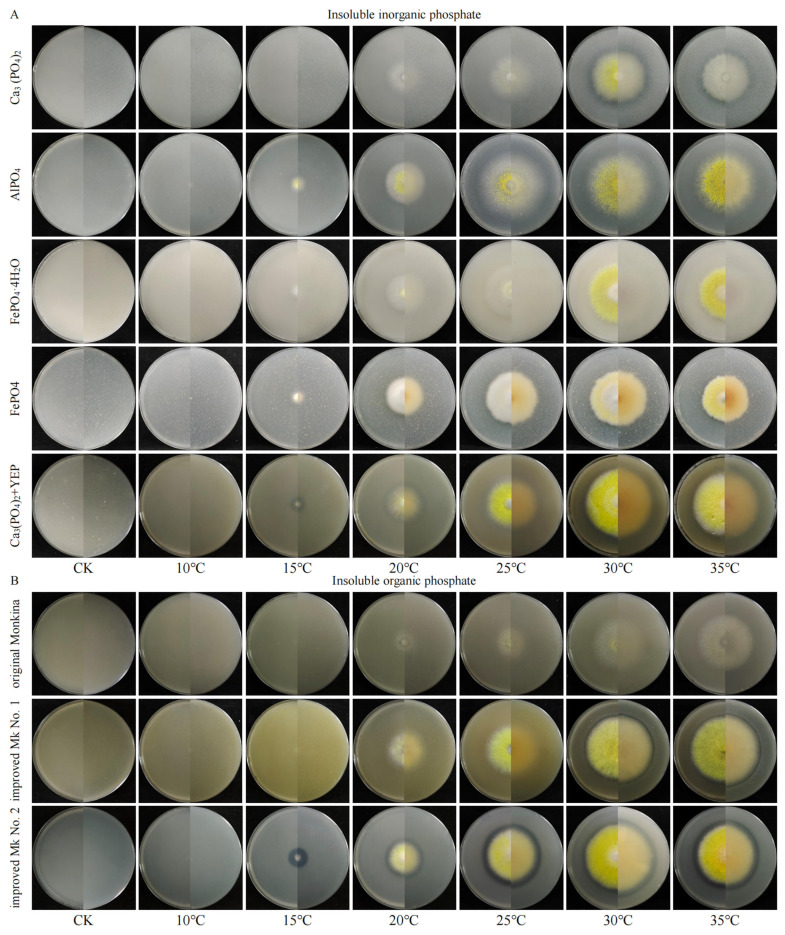
Preliminary qualitative testing of insoluble phosphate decomposition ability in *T. nanjingensis*. Note: From left to right, the uninoculated group (CK) and the inoculated group (10 °C, 15 °C, 20 °C, 25 °C, 30 °C and 35 °C) were shown. (**A**) Decomposition of insoluble inorganic phosphate by *T. nanjingensis* cultured for 7 days. From top to bottom was NBRIP medium supplemented with Ca_3_(PO_4_)_2_, AlPO_4_, FePO_4_·4H_2_O and FePO_4_, as well as Ca_3_(PO_4_)_2_ and YEP (0.5 g/L), respectively. (**B**) Decomposition of insoluble organic phosphate by *T. nanjingensis* cultured for 7 days. From top to bottom was was original Monkina medium (lecithin 0.6 g/L), improved Monkina medium No. 1 (lecithin 5 g/L), improved Monkina medium No. 2 (lecithin 0.2 g, YEP 0.5 g/L), respectively. In this Figure, Monkina was abbreviated as MK.

**Figure 2 microorganisms-13-01574-f002:**
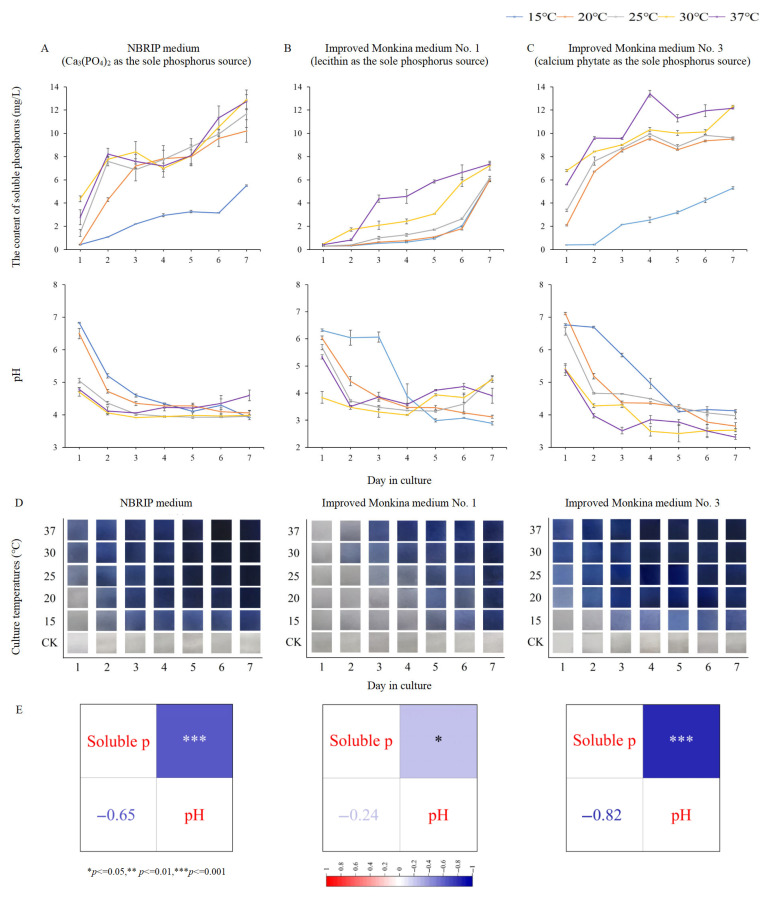
Phosphate-solubilizing characteristics and dynamic pH changes of *T. nanjingensis* under different culture temperatures and culture periods. (**A**) Changes in soluble phosphorus and pH in insoluble inorganic phosphate NBRIP medium (Ca_3_(PO_4_)_2_) at different culture temperatures with time after the addition of *T. nanjingensis*. (**B**) In insoluble organic phosphate improved Monkina medium No. 1 (lecithin) and (**C**) improved Monkina medium No. 3 (calcium phytate). (**D**) In the process of using the molybdenum–antimony spectrophotometry method (Mo-Sb-Vc) to determine the amount of soluble phosphorus dissolved by *T. nanjingensis* at different culture temperatures and periods, changes in color in the corresponding trace 96-well plate cuvette were observed. CK denotes the blank group without inoculation. (**E**) The correlation between the soluble phosphorus content and pH value in 3 types of insoluble phosphate liquid medium after adding *T. nanjingensis*. Note: *, **, *** denote the different levels of stepwise statistically significant differences that may occur in (**E**) when calculating significance.

**Figure 3 microorganisms-13-01574-f003:**
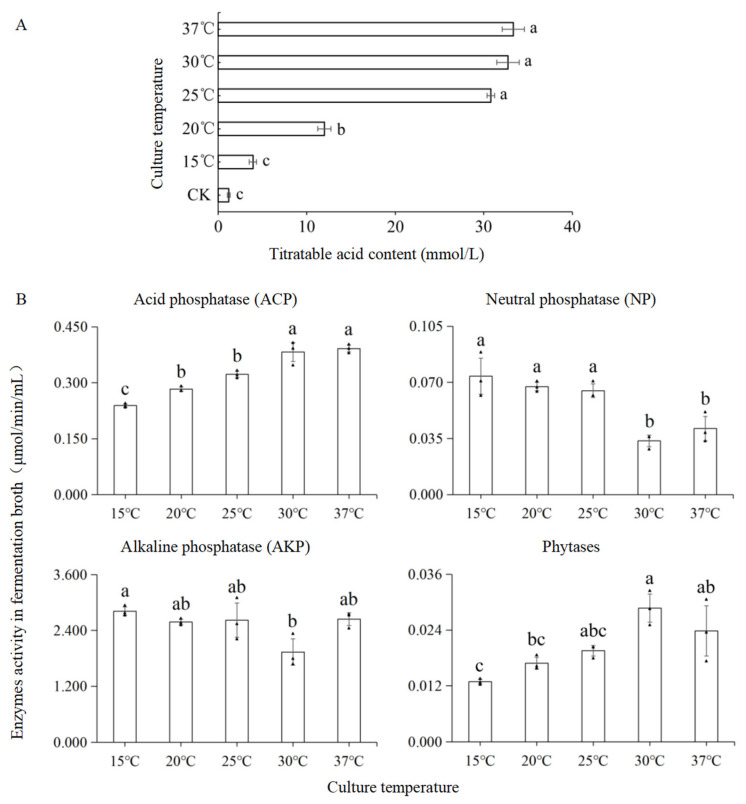
Production of titratable acid and activity of phosphate-solubilizing enzymes in fermentation broth of *T. nanjingensis*. (**A**) Titratable acid production of *T. nanjingensis* at different fermentation temperatures. (**B**) Activity of 4 phosphate-solubilizing enzymes in *T. nanjingensis* at different fermentation temperatures. Note: Significant (*p* < 0.05) differences between different fermentation temperatures are identified by lowercase letters (a, b, c), respectively.

**Figure 4 microorganisms-13-01574-f004:**
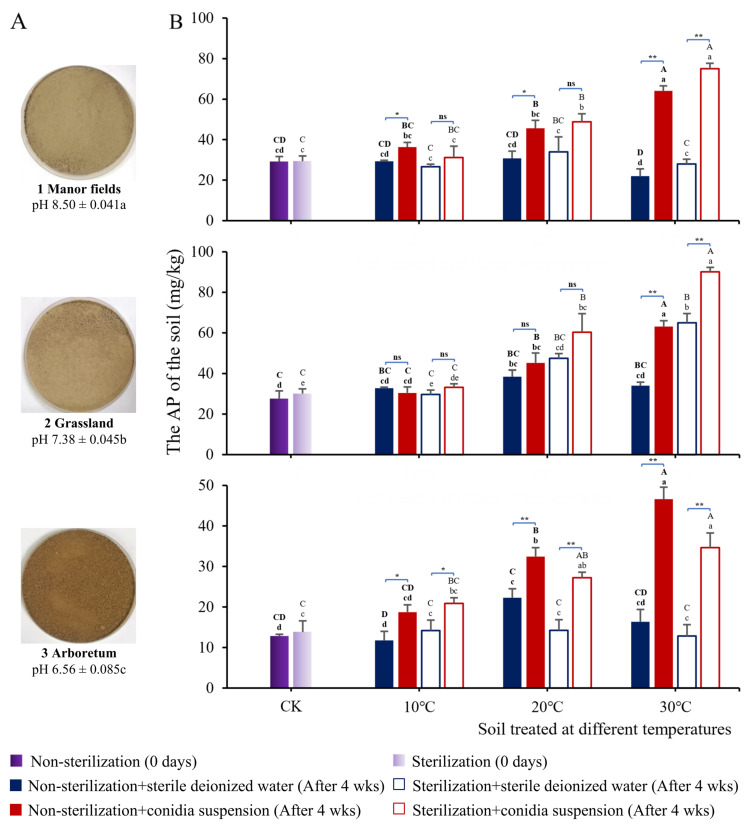
Effects of *T. nanjingensis* on 3 different soils. (**A**) Three soils with different pH values from 3 sampling locations; (**B**) AP of soils in the non-sterilization and sterilization treatment groups at 3 temperature conditions for 4 weeks after the addition of conidia suspensions from *T. nanjingensis*. Note: ANOVA and Tukey’s multiple comparisons were employed to assess differences. Significant (*p* < 0.05) and highly significant (*p* < 0.01) differences between different treatment groups are identified by lowercase letters (a, b, c, d, etc.) and uppercase letters (A, B, C, D), respectively. In particular, for AP (see Section 3.5), an additional two-samples T test was introduced (* for *p* < 0.05, ** for *p* < 0.01). The indicated letters used for the non-sterilization group data are bolded, and the original font is used for the sterilization group. “ns” is the abbreviation for “no significant difference”.

**Figure 5 microorganisms-13-01574-f005:**
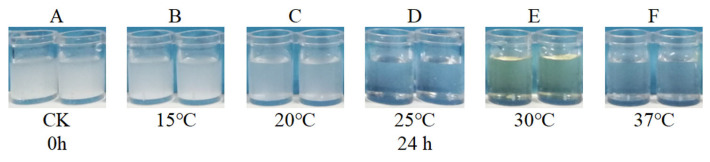
The change in the turbidity degree after the addition of *T. nanjingensis* in the original Monkina liquid medium. (**A**) The original Monkina liquid medium when freshly prepared (0 h). (**B**–**F**) From left to right, 24 h after inoculation with *T. nanjingensis* under 15 °C, 20 °C, 25 °C, 30 °C, and 37 °C.

**Table 1 microorganisms-13-01574-t001:** The content of phosphates and phosphorus in the phosphate-solubilizing capability detection media.

Medium Name	Insoluble Phosphates/Molecular Weight	CAS No.	Phosphate Content (g/L)	Proportion of Phosphorus in Phosphate	Phosphorus Content (g/L)
Original NBRIP	Ca_3_(PO_4_)_2_, 310.177	7758-87-4	5.00	61.948/310.177 = 1/5.007	0.999 ≈ 1.0
	AlPO_4_, 121.953	7784-30-7	3.94	30.974/121.953 = 1/3.937	1.0
	FePO_4_·4H_2_O, 222.877	14940-41-1	7.20	30.974/222.877 = 1/7.196	1.0
	FePO_4_, 150.816	10045-86-0	4.87	30.974/150.816 = 1/4.869	1.0
Original Monkina	Lecithin C_42_H_80_NO_8_P, 758.060	97281-47-5	0.60	30.974/758.060 = 1/24.474	0.025
Improved MK No. 1	Lecithin C_42_H_80_NO_8_P, 758.060	97281-47-5	5.00	30.974/758.060 = 1/24.474	0.204
Improved MK No. 2	Lecithin C_42_H_80_NO_8_P, 758.060	97281-47-5	0.20	30.974/758.060 = 1/24.474	0.008
Improved MK No. 3	Calcium phytate C_6_H_6_Ca_6_O_24_P_6_, 888.408	7776-28-5	5.00	185.844/888.408 = 1/4.780	1.046

Note: Monkina is abbreviated as MK.

**Table 2 microorganisms-13-01574-t002:** Comparison of chemical and physical properties of soil samples from different collection points.

Soil Sampling Point		1 Manor Fields, Xuzhou, Jiangsu Province	2 Grasslands, Nanjing, Jiangsu Province	3 Arboretum, Nanjing, Jiangsu Province
Latitude and Longitude of Sampling Point		34°43′22″ N, 116°56′49″ E	32°4′43″ N, 118°48′37″ E	32°4′50″ N, 118°49′13″ E
Chemical and physical properties	AN (mg/kg)	37.607	251.952	71.620
	TP (g/kg)	0.813 ± 0.009 b	0.880 ± 0.005 a	0.346 ± 0.014 c
	AP (mg/kg)	29.041 ± 2.607 a	27.613 ± 3.833 a	12.769 ± 0.503 b
	AK (mg/kg)	245.422 ± 7.213 b	459.365 ± 11.804 a	90.583 ± 5.437 c
	HM (g/kg)	7.888	12.102	11.594
	pH	8.50 ± 0.041 a	7.38 ± 0.045 b	6.56 ± 0.085 c
	EC (μs/cm)	914.80 ± 4.224 a	188.53 ± 2.076 b	23.84 ± 0.735 c

Note: Significant (*p* < 0.05) differences between different collection points are identified by lowercase letters (a, b, c), respectively.

**Table 3 microorganisms-13-01574-t003:** Genes associated with phosphate decomposition and environmental adaptability found in *T. nanjingensis*.

		Type of Gene	Number of Genes (or Items)
Phosphate decomposition	Phosphate solubilizing	organic acid	9
		acid phosphatase	29
		alkaline phosphatase	28
		phytase	7
		phosphonatase	3
		C-P lyase	35
	Phosphate- related	phospholipase	90
		dolichyldiphosphatase	1
		pyrophosphatase	18
		diphosphatase	15
		phosphoesterase	36
		phosphodiesterase	37
Temperature-related		Low temperature	8
		Cold	1
		High temperature	1
		Temperature-dependent	1
Salt-related		Salt responsiveness or salt tolerance	4
Stress resistance-related		1 Antibiotic resistance	541 items
		2 Antifungal agent resistance	
		3 Drug resistance	
		4 Pesticide resistance	
		5 UV radiation resistance	
		6 Disease resistance	
		7 Metal resistance	
		8 Acid resistance	
		9 Natural resistance	
		10 Other chemical resistance	
Genes that interact with plants and microbes	Siderophores	Siderophore	20
		Pyoverdine	5
		Catechol- type	3
		Hydroxamate-type	1
	Hormones	Auxin	2
		Gibberellin	2
		Indole-3-acetic acid	4
		1-Aminocyclopropane-1-carboxylate deaminase (ACC deaminase)	6
	Other	Salicylic acid/salicylate	71 items
		Polyamine	144 items

## Data Availability

The genomic information of *T. nanjingensis* has been uploaded to the NCBI (https://www.ncbi.nlm.nih.gov/nuccore/JARFPJ000000000.1/, accessed on 5 June 2025). The BioProject ID was PRJNA941179, the TaxID for genome sequencing was 2916467, and the Locus Tag Prefix was P0O16. The accession number of the BioSample was SAMN33591157. The whole-genome shotgun data have been deposited at DDBJ/ENA/GenBank under the accession number JARFPJ000000000; the version described in this paper was JARFPJ010000000.

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
