# Peer review of "Systematic Investigation of Phosphate Decomposition and Soil Fertility Modulation by the Filamentous Fungus Talaromyces nanjingensis"

_microorganisms, 2025, doi:10.3390/microorganisms13071574_

Round 1

Reviewer 1 Report

Comments and Suggestions for Authors

Manuscripr entitled Systematic Investigation of Phosphate Decomposition and Soil 2
Fertility Modulation by the Filamentous Fungus Talaromyces >
nanjingensis  presents useful information on Phosphate-solubilizing microbes in soil, which  convert insoluble phosphate into plant-available soluble phosphorus.

The paper systematically presents qualitative and quantitative techniques for assess phosphate decomposing capabilities of microbes. Additionally,  presenting two optimized media for these microorganisms.

Minor details required:

--Please, correct the abbreviations of unities:

  --line21: the secretion of organic acids, including gluconic acid (6.10 gL-1, oxalic
acid (0.93 gL-1), and malonic acid (0.17 g/L), 

secretion of organic acids, including gluconic acid (6.10 g/L), oxalic
acid (0.93 g/L), and malonic acid (0.17 g/L), 

to use : gL-1

UmL-1

along the text.

line 108: t (1 g/L)--  gL-1

Figure2:  check for : soluble not souble

to check for italics in scientific names

Also check table1

Also add any hypothesis and future studies.

the paper is well presented containing 3 tables and 4 figures

the paper presents a new detection culture media  for the rapid screening of phosphate
solubilizing microorganisms 

Reviewer 2 Report

Comments and Suggestions for Authors

In the scientific article the authors investigated the phosphate-solubilizing ability of T. nanjingensis under different temperature gradients and its applicability in soils with different pH values, and the molecular basis of phosphate decomposition. The data described are valuable and should be published. The manuscript was written in a friendly way, however  some aspect should be improved before publication.

Introduction

  1. The authors did not highlight the novelty of the MS compared to other published work. I can not identify any novel approach in the MS.
  2. There is a lack of a hypothesis or hypotheses.

Materials and methods

  1. Generally, the materials and methods are well described, however one of the important aspect in the MS – genetic analyses were poorly written and should be improved.
  2. The authors wrote “Second- and third-generation genome sequencing techniques was used for T. nanjingensis” (line 226) . Please describe what platform did you use to perform sequencing. Please describe in detail the procedure e.g. DNA isolation, kits etc.
  3. How did you classify the genes “associated with phosphate decomposition and environmental adaptability found in T. nanjingensis” – describe in detail.

Conclusions

  1. Please describe the limitations of the conducted experiments.
  2. Please describe the challenges and future studies that should be conducted.

Round 2

Reviewer 2 Report

Comments and Suggestions for Authors

The manuscript could be accepted in the revised form.